# Thermodynamic Properties of Electrolyte Solutions, Derived from Fluctuation Correlations: A Methodological Review

**Dimiter N. Petsev**

Department of Chemical and Biological Engineering and Center for Micro-Engineered Materials,
University of New Mexico, Albuquerque, NM 87131, USA; dimiter@unm.edu

**Abstract:** This article presents a review of an approach for studying solution thermodynamics, which is based the on hydrodynamic fluctuation correlations analysis method suggested by Landau and Lifshitz. We show that the method is very general, and its applicability goes beyond hydrodynamics. It starts with examining the entropy production and fluctuating transport fluxes, which are related to concentration fluctuations and molecular interactions. The approach can be successfully applied to compute a wide range of thermodynamic properties such as the osmotic pressure (i.e., equation of state) and provides information about the interactions between the dissolved species. Using dilute electrolyte solutions as a case study, we reproduce results from the Debye and Huckel theory while starting from a very different physical perspective.

**Keywords:** fluctuations; electrolyte solutions; solution thermodynamics

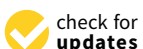



## 1. Introduction

This paper offers an overview of the Landau and Lifshitz approach for the analysis of hydrodynamic fluctuations [1], and its potential to be used as a tool for obtaining meaningful and important physical results. The starting point for the analysis is the fundamental law of entropy production, and therefore, its foundations are quite general. The Landau and Lifshitz theory of hydrodynamic fluctuations is very helpful in studying stochastic properties in viscous liquids. An intriguing example is the derivation of the Stokes–Einstein diffusion coefficient by integrating the fluctuating viscous stresses acting on the surface of a spherical particle, as demonstrated by Zwanzig [2] and later by Fox and Uhlenbeck [3]. The general approach can be extended beyond fluid momentum and heat transfer. It was used to offer a possible explanation for the attractive interactions between like-charged particles in colloid crystals [4], the surface forces in thin liquid films due to surface capillary waves [5], or ionic concentration-fluctuation correlations in electric double layers [6].

The Landau and Lifshitz method can be adapted to examine the properties of solutions, including electrolytes. This was demonstrated by Donev et al. [7] and Peraud et al. [8], who studied the fluctuations in such systems in detail. Their main focus was on the non-equilibrium, gradient-driven transport in fluid mixtures and hence requires a numerical approach. Still, they offered results for the equilibrium properties of solutions such as structure factors and the corresponding radial distribution functions in the Debye and Huckel limit. The focus of the present review is on the general analytical methodology that allows for the derivation of thermodynamic equilibrium properties of electrolyte solutions such as osmotic pressure, radial distribution functions, and interaction energies between ions (potentials of mean force). The effect of the electrostatics enters the analysis via the Nernst–Planck equations [9] for the fluxes in combination with the Poisson equation that relates the electrostatic potential to the charge in the fluid [10]. We provide a detailed pathway for deriving the Debye and Huckel mean field theory of strong electrolytes [11,12]), starting from the general relationship for the entropy production. All assumptions are

clearly outlined, which helps to better understand the limitations of the Debye and Huckel model. This analysis is less general than the one offered by Peraud et al. [8], as it does not include non-equilibrium effects. Abandoning some of the simplifications (dilute solutions, small fluctuation amplitudes, solvent structural molecular contributions, etc.) may in principle lead to a more general theory. The complexity of the calculations, however, would increase and obtaining analytical results may again become impossible.

## 2. Theoretical Approach

In this section, we present a brief overview of the Landau and Lifshitz description of fluctuating hydrodynamics [1] and outline the application of the general method to other transport processes such as mass diffusion. The latter is necessary in order to examine the concentration fluctuations, which will next be used to determine the thermodynamic properties of solutions.

### 2.1. Entropy Production and Fluxes

We start with examining the entropy production rate in the solution, $S$ (i.e., its change with time, $t$), which is given by [1,13]

$$\frac{dS}{dt} = \frac{d_i S}{dt} + \frac{d_e S}{dt}. \tag{1}$$

The first term on the right-hand side is due to irreversible processes inside the thermodynamic system and according to the second law of thermodynamics is always positive, or $d_i S/dt \geq 0$. The second term accounts for the entropy change that results from exchanges of energy and matter with the systems surroundings. It is the irreversible part of the total entropy charge that is relevant to the current discussion. In the case of a solution, it reads [1,13]

$$\frac{1}{k_B}\frac{d_i S}{dt} = \frac{\dot{S}}{k_B} = -\int_{\Delta V} dV \left( \mathbf{i} \cdot \frac{\nabla \mu}{k_B T} \right), \tag{2}$$

where $\mathbf{i}$ is the diffusion flux in $\mathrm{kg\,m^{-2}s^{-1}}$, $\mu$ is the chemical potential, and $\Delta V$ is some volume within the solution. If the volume $\Delta V$ is small, Equation (2) becomes

$$\frac{\dot{S}}{k_B} \approx -\mathbf{i} \cdot \frac{\nabla \mu}{k_B T} \Delta V, \tag{3}$$

Following Landau and Lifshitz [1], we can write Equation (2) (or Equation (3)) in the form

$$\dot{S} = \int_{\Delta V} dV \left( \sum_\eta X_\eta \dot{x}_\eta \right) \approx \sum_\eta X_\eta \dot{x}_\eta \delta V. \tag{4}$$

The quantities $X_\eta$ and $\dot{x}_\eta$ are the generalized thermodynamic forces and fluxes, respectively [1,13,14]. Comparing Equation (4) to (2) leads to

$$X_\eta = \frac{1}{k_B T}\frac{\partial \mu}{\partial r_\eta}, \quad \dot{x}_\eta = i_\eta \tag{5}$$

where we have set $\mu = \mu_1/m_1 - \mu_2/m_2$. The index $\eta = 1,2,3$ accounts for the three spatial coordinates $r_1, r_2, r_3$. The total flux can then be expressed as

$$\mathbf{i} = -\alpha \nabla \mu + \delta \mathbf{i}, \tag{6}$$

where $\delta \mathbf{i}$ is the fluctuating part of the flux $\mathbf{i}$.

### 2.2. Correlation of the Fluctuating Diffusion Fluxes

The generalized fluxes can formally be expressed by [1,14]

$$\dot{x}_\eta = -\sum_\zeta \gamma_{\eta\zeta} X_\zeta + y_\eta. \tag{7}$$

The indices $\eta$ and $\zeta$ correspond to various transport processes in all three spatial directions that may be present in the solution, where the fluctuating fluxes correlations are given by [14,15]

$$\langle y_\eta(t_1) y_\zeta(t_2) \rangle = (\gamma_{\eta\zeta} + \gamma_{\zeta\eta}) \delta(t_1 - t_2). \tag{8}$$

The angular brackets $\langle \ldots \rangle$ denote ensemble averaging [12,14]. Note that $\langle y_\eta(t_1) \rangle = 0$. We can write the $\eta$-component of Equation (6) in the form

$$i_\eta = -\sum_\zeta \left( \frac{\alpha k_B T}{\Delta V} \delta_{\eta\zeta} \right) \left( \frac{1}{k_B T} \frac{\partial \mu}{\partial r_\zeta} \Delta V \right) + \delta i_\eta \tag{9}$$

A comparison of Equation (9) with Equation (7) leads to the following expressions for the coefficient $\gamma_{\eta\zeta}$ and the fluctuating flux $\delta i_\eta$

$$\gamma_{\eta\zeta} = \frac{\alpha k_B T}{\Delta V} \delta_{\eta\zeta}, \ y_\eta = \delta i_\eta, \tag{10}$$

where again $\delta i_\eta$ denotes the fluctuation of the $\eta$ component of the mass flux. The symbol $\delta_{\eta\zeta}$ is the Kronecker delta [16]. Combining Equations (7)–(10) and noting that $\gamma_{\eta\zeta} = \gamma_{\zeta\eta}$ [15] allows obtaining

$$\langle \delta i_\eta(\mathbf{r}_1, t_1) \delta i_\zeta(\mathbf{r}_2, t_2) \rangle = \frac{2\alpha k_B T}{\Delta V} \delta_{\eta\zeta} \delta(t_1 - t_2). \tag{11}$$

Taking the limit $\Delta V \to 0$ transforms Equation (11) into

$$\langle \delta i_\eta(\mathbf{r}_1, t_1) \delta i_\zeta(\mathbf{r}_2, t_2) \rangle = 2\alpha k_B T \delta_{\eta\zeta} \delta(t_1 - t_2) \delta(\mathbf{r}_1 - \mathbf{r}_2). \tag{12}$$

Let us consider a two-component system where component 1 is the solute and component 2 is the solvent. The fundamental thermodynamic equation for the internal energy $E$ reads [12,14]

$$dE = TdS - pdV + \mu_1 dN_1 + \mu_2 dN_2 \tag{13}$$

where $T$ is temperature, $p$ is pressure, $V$ is volume and $N_1$ and $N_2$ are the number of molecules of type 1 and 2. The chemical potentials of the solute and solvent read

$$\mu_1 = \mu_1^0 + k_B T \ln N_1, \text{ and } \mu_2 = \mu_2^0 + k_B T \ln N_2. \tag{14}$$

If the corresponding molecular masses are $m_1$ and $m_2$, then we can write total mass as

$$M_t = N_1 m_1 + N_2 m_2 \tag{15}$$

Combining Equations (13) and (15), and introducing a new variable $c = N_1 m_1$, leads to

$$dE = TdS - pdV + \left( \frac{\mu_1}{m_1} - \frac{\mu_2}{m_2} \right) dc. \tag{16}$$

The diffusion coefficient for component 1 is defined as

$$D = \frac{\alpha}{\rho} \left( \frac{\partial \mu}{\partial n_1} \right) = \frac{\alpha V}{m_1} \frac{\partial \mu}{\partial N_1}, \tag{17}$$

where $\rho$ is the solution density (see Equations (13) and (16))

$$\rho = \frac{N_1 m_1 + N_2 m_2}{V} \tag{18}$$

and $n_1$ is the mass fraction of component 1, or

$$n_1 = \frac{N_1 m_1}{N_1 m_1 + N_2 m_2} \tag{19}$$

Assuming $n_1 \ll 1$ (or $N_1 \ll N_t$, where $N_t$ is the total number of molecules), and taking the derivative of the combined chemical potential $\mu = \mu_1/m_1 - \mu_2/m_2$ with respect to $N_1$ leads to

$$D = \frac{\alpha V}{m_1} \frac{\partial}{\partial N_1} \left( \frac{\mu_1}{m_1} - \frac{\mu_2}{m_2} \right) = \frac{\alpha}{\rho} \left( \frac{k_B T}{m_1 N_1} + \frac{k_B T}{m_2 (N_t - N_1)} \right) \approx \frac{\alpha k_B T}{m_1^2 c_1} \tag{20}$$

where $c_1 = N_1/V$ is the number concentration of component 1. Solving for $\alpha$ results in

$$\alpha = \frac{m_1^2 c_1 D}{k_B T}. \tag{21}$$

It is helpful to introduce fluctuation fluxes per unit mass as $\delta \mathbf{j} = \delta \mathbf{i}/m_1$ and $\delta j_\eta = \delta i_\eta/m_1$. Replacing Equation (21) in Equation (12) yields

$$\langle \delta j_\eta(\mathbf{r}_1, t_1) \delta j_\zeta(\mathbf{r}_2, t_2) \rangle = 2cD\delta_{\eta\zeta}\delta(t_1 - t_2)\delta(\mathbf{r}_1 - \mathbf{r}_2) \tag{22}$$

or in tensor form

$$\langle \delta \mathbf{j}(\mathbf{r}_1, t_1) \delta \mathbf{j}(\mathbf{r}_2, t_2) \rangle = 2cD\mathbf{I}\delta(t_1 - t_2)\delta(\mathbf{r}_1 - \mathbf{r}_2), \tag{23}$$

with $\mathbf{I}$ being the unit tensor with elements $\delta_{\eta\zeta}$. Equations (22) and (23) imply that only the fluxes of the same species are correlated. The correlation of fluxes of different species is zero.

## 3. Fluctuation Correlations and Thermodynamics of Binary Symmetric Electrolyte Solutions

A binary electrolyte solution is actually a ternary system that consists of positive ions, negative ions, and solvent. However, if the concentrations of both positive and negative ions ($c_+$ and $c_-$) is low in comparison to that of the solvent $c$, then each ionic species presents an ideal solution, and all results from the previous section are applicable to each of them. Our analysis focuses on symmetric electrolytes where the ions carry the same charge numbers, $z$, but with opposite signs. The presence of the charges in the solution, however, creates an additional difficulty associated with the electric fields and electrostatic interactions between the charged components. The celebrated theory of Debye and Huckel [11] was the first successful attempt to overcome these difficulties and is still included in most texts that are dedicated to the physics and chemistry of electrolyte solutions. The fluctuation correlation analysis, described here, offers a different starting point to approach the same problem. It reveals some important physical insights and provides strategies for solving similar problems in a relatively straightforward way.

### 3.1. Balance Equations

We start with balancing the mass of dissolved electrolytes as well as the potential in the solution. The mass balance Nernst–Planck equations read

$$
\begin{aligned}
\frac{\partial c_+}{\partial t} &= -\nabla \cdot \mathbf{j}_+ + ze\beta_+ \nabla \cdot (c_+ \nabla \psi) \\
\frac{\partial c_-}{\partial t} &= -\nabla \cdot \mathbf{j}_- - ze\beta_- \nabla \cdot (c_- \nabla \psi).
\end{aligned}
\tag{24}
$$

The coefficients $\beta_+$ and $\beta_-$ are the hydrodynamic mobilities of the positive and negative ions, while $e$ is the elementary charge. The electrostatic potential $\psi$ is related to the charge density $\rho_e = ze(c_+ - c_-)$ by means of the Poisson equation [10]

$$
\nabla^2 \psi = -\frac{\rho_e}{\varepsilon \varepsilon_0},
\tag{25}
$$

where $\varepsilon_0$ is the dielectric constant in vacuo, and $\varepsilon$ is the relative dielectric permittivity.

In order to account for the fluctuations in the solution, we define the fluxes, the concentrations, and the local potential as

$$
\begin{aligned}
\mathbf{j}_+ &= -D_+ \nabla c_+ + \delta \mathbf{j}_+ \\
\mathbf{j}_- &= -D_- \nabla c_- + \delta \mathbf{j}_-,
\end{aligned}
\tag{26}
$$

$$
\begin{aligned}
c_+ &= c + \delta c_+ \\
c_- &= c + \delta c_-,
\end{aligned}
\tag{27}
$$

and

$$
\psi = 0 + \delta \psi,
\tag{28}
$$

where $D_+ = k_B T / \beta_+$ and $D_- = k_B T / \beta_-$ are the diffusion coefficients for the positive and negative ions, $\delta c_+$ and $\delta c_-$ are the concentration fluctuations for the positive and negative ions, and $c$ is the uniform average concentration in units of number per volume. The resultant local electrostatic potential fluctuation is $\delta \psi$.

The balance equations for all fluctuating quantities become

$$
\begin{aligned}
\frac{\partial \delta c_+}{\partial t} &= \nabla^2 \delta c_+ - \nabla \delta \cdot \mathbf{j}_+ + ze\beta_+ c \nabla^2 \delta \psi \\
\frac{\partial \delta c_-}{\partial t} &= \nabla^2 \delta c_- - \nabla \delta \cdot \mathbf{j}_- - ze\beta_- c \nabla^2 \delta \psi.
\end{aligned}
\tag{29}
$$

$$
\nabla^2 \delta \psi = -\frac{\delta \rho_e}{\varepsilon \varepsilon_0},
\tag{30}
$$

The charge fluctuation is

$$
\delta \rho_e = ze(\delta c_+ - \delta c_-) = \delta \rho_+ + \delta \rho_-,
\tag{31}
$$

where

$$
\delta \rho_+ = ze\delta c_+, \text{ and } \delta \rho_- = -ze\delta c_-.
\tag{32}
$$

Rearranging the above equations yields

$$
\begin{aligned}
\frac{\partial \delta \rho_+}{\partial t} &= D_+ \left( \nabla^2 \delta \rho_+ - \frac{\kappa^2}{2} \delta \rho_e \right) - ze\nabla \cdot \delta \mathbf{j}_+ \\
\frac{\partial \delta \rho_-}{\partial t} &= D_- \left( \nabla^2 \delta \rho_- - \frac{\kappa^2}{2} \delta \rho_e \right) + ze\nabla \cdot \delta \mathbf{j}_-, \\
\kappa^2 &= \frac{2z^2 e^2 c}{\varepsilon \varepsilon_0 k_B T} = \frac{2z^2 e^2 c\beta_+}{\varepsilon \varepsilon_0 D_+} = \frac{2z^2 e^2 c\beta_-}{\varepsilon \varepsilon_0 D_-}.
\end{aligned}
\tag{33}
$$

The parameter $\kappa$ is the Debye screening parameter (inverse length) [11], which is a measure of potential magnitude reduction with distance around an ion in the solution. A significant simplification of Equation (33) can be achieved by setting $D_+ = D_- = D$, and noting that for symmetric electrolytes, the overall concentrations are $c_+ = c_- = c$. This seems like a very strong limitation, but it does not affect the majority of the final results. Combining the first two equations in (33) leads to an equation for the charge density fluctuation

$$\frac{\partial \delta \rho_e}{\partial t} = D\left(\nabla^2 \delta \rho_e - \kappa^2 \delta \rho_e\right) - ze\nabla \cdot (\delta \mathbf{j}_+ - \delta \mathbf{j}_-). \tag{34}$$

### 3.2. Charge Fluctuation Correlations and Correlation Energy

We will use the Fourier transform technique [16] to express the charge and flux fluctuations as

$$\delta \hat{\rho}_e(\mathbf{k}, \omega) = \frac{1}{(2\pi)^2} \int_{-\infty}^{\infty} d\mathbf{r} \int_{-\infty}^{\infty} dt \rho_e(\mathbf{r}, t) e^{i(\omega t - \mathbf{k} \cdot \mathbf{r})} \tag{35}$$

and

$$\delta \hat{\mathbf{j}}(\mathbf{k}, \omega) = \frac{1}{(2\pi)^2} \int_{-\infty}^{\infty} d\mathbf{r} \int_{-\infty}^{\infty} dt \delta \mathbf{j}(\mathbf{r}, t) e^{i(\omega t - \mathbf{k} \cdot \mathbf{r})}. \tag{36}$$

The inverse transforms are

$$\delta \rho_e(\mathbf{r}, t) = \frac{1}{(2\pi)^2} \int_{-\infty}^{\infty} d\mathbf{k} \int_{-\infty}^{\infty} d\omega \hat{\rho}_e(\mathbf{k}, \omega) e^{-i(\omega t - \mathbf{k} \cdot \mathbf{r})} \tag{37}$$

and

$$\delta \mathbf{j}(\mathbf{r}, t) = \frac{1}{(2\pi)^2} \int_{-\infty}^{\infty} d\mathbf{k} \int_{-\infty}^{\infty} d\omega \delta \hat{\mathbf{j}}(\mathbf{k}, \omega) e^{-i(\omega t - \mathbf{k} \cdot \mathbf{r})} \tag{38}$$

where $\mathbf{k}$ is the wavevector and $\omega$ is the frequency.

The ionic fluxes and their correlations are of special interest. For low ionic concentrations, the diffusion flux of each component does not depend on any variable pertinent to the other solutes (see Equations (20)). Hence, the correlations between the fluxes for different species is zero, or

$$\begin{aligned}
\langle \delta \mathbf{j}_+(\mathbf{r}_1, t_1) \delta \mathbf{j}_+(\mathbf{r}_2, t_2) \rangle &= 2cD\mathbf{I}\delta(t_1 - t_2)\delta(\mathbf{r}_1 - \mathbf{r}_2), \\
\langle \delta \mathbf{j}_-(\mathbf{r}_1, t_1) \delta \mathbf{j}_-(\mathbf{r}_2, t_2) \rangle &= 2cD\mathbf{I}\delta(t_1 - t_2)\delta(\mathbf{r}_1 - \mathbf{r}_2), \\
\langle \delta \mathbf{j}_+(\mathbf{r}_1, t_1) \delta \mathbf{j}_-(\mathbf{r}_2, t_2) \rangle &= \langle \delta \mathbf{j}_-(\mathbf{r}_1, t_1) \delta \mathbf{j}_+(\mathbf{r}_2, t_2) \rangle = 0.
\end{aligned} \tag{39}$$

Equations (38) and (39) allow for the derivation of the important relationship (see Appendix A)

$$\langle \delta \hat{\mathbf{j}}(\mathbf{k}_1, \omega_1) \delta \hat{\mathbf{j}}(\mathbf{k}_2, \omega_2) \rangle = 2c\mathbf{I}D\delta(\omega_1 + \omega_2)\delta(\mathbf{k}_1 + \mathbf{k}_2). \tag{40}$$

Note that the right-hand sides for the first two equations in (40) are the same irrespective of whether the positive or negative ionic fluxes are correlated.

Equations (35)–(38) allow Fourier transforming the whole Equation (34)

$$i\omega \delta \hat{\rho}_e(\mathbf{k}, \omega) = -D(k^2 + \kappa^2)\delta \hat{\rho}_e(\mathbf{k}, \omega) + ize\mathbf{k} \cdot [\delta \hat{\mathbf{j}}_+(\mathbf{k}, \omega) - \delta \hat{\mathbf{j}}_-(\mathbf{k}, \omega)]. \tag{41}$$

Solving for the charge density $\delta \hat{\rho}_e(\mathbf{k}, \omega)$ yields

$$\delta \hat{\rho}_e(\mathbf{k}, \omega) = \frac{i\mathbf{k}ze[\delta \hat{\mathbf{j}}_+(\mathbf{k}, \omega) - \delta \hat{\mathbf{j}}_-(\mathbf{k}, \omega)]}{i\omega + D(k^2 + \kappa^2)}. \tag{42}$$

The charge correlation fluctuation correlation is

$$\langle \delta\hat{\rho}_e(\mathbf{k}_1,\omega_1)\delta\hat{\rho}_e(\mathbf{k}_2,\omega_2) \rangle =$$
$$-\frac{\mathbf{k}_1 \cdot (ze)^2[\langle \delta\hat{\mathbf{j}}_+(\mathbf{k}_1,\omega_1)\delta\hat{\mathbf{j}}_+(\mathbf{k}_2,\omega_2) \rangle + \langle \delta\hat{\mathbf{j}}_-(\mathbf{k}_1,\omega_1)\delta\hat{\mathbf{j}}_-(\mathbf{k}_2,\omega_2) \rangle] \cdot \mathbf{k}_2}{[i\omega_1 + D(k_1^2 + \kappa^2)][i\omega_2 + D(k_2^2 + \kappa^2)]}. \tag{43}$$

Using Equations (40) and (43), the identity $\mathbf{k}_1 \cdot \mathbf{I} \cdot \mathbf{k}_2 = \mathbf{k}_1 \cdot \mathbf{k}_2$ leads to

$$\langle \delta\hat{\rho}_e(\mathbf{k}_1,\omega_1)\delta\hat{\rho}_e(\mathbf{k}_2,\omega_2) \rangle = -\frac{4(ze)^2cD\mathbf{k}_1 \cdot \mathbf{k}_2\delta(\omega_1+\omega_2)\delta(\mathbf{k}_1+\mathbf{k}_2)}{[i\omega_1 + D(k_1^2 + \kappa^2)][i\omega_2 + D(k_2^2 + \kappa^2)]}. \tag{44}$$

The electrostatic energy per unit volume of the solution is (see Figure 1)

$$\frac{E_e}{V} = \frac{1}{2}\langle \delta\psi(\mathbf{r}_1,t_1)\delta\rho_e(\mathbf{r}_2,t_2) \rangle. \tag{45}$$

Using the Fourier transform of Equation (30)

$$-k_1^2\delta\hat{\psi} = -\frac{\delta\hat{\rho}_e}{\varepsilon\varepsilon_0} \text{ or } \delta\hat{\psi} = \frac{1}{k_1^2}\frac{\delta\hat{\rho}_e}{\varepsilon\varepsilon_0} \tag{46}$$

Then, the correlation of the potential and charge fluctuations in Equation (45) becomes (see also Equation (43)).

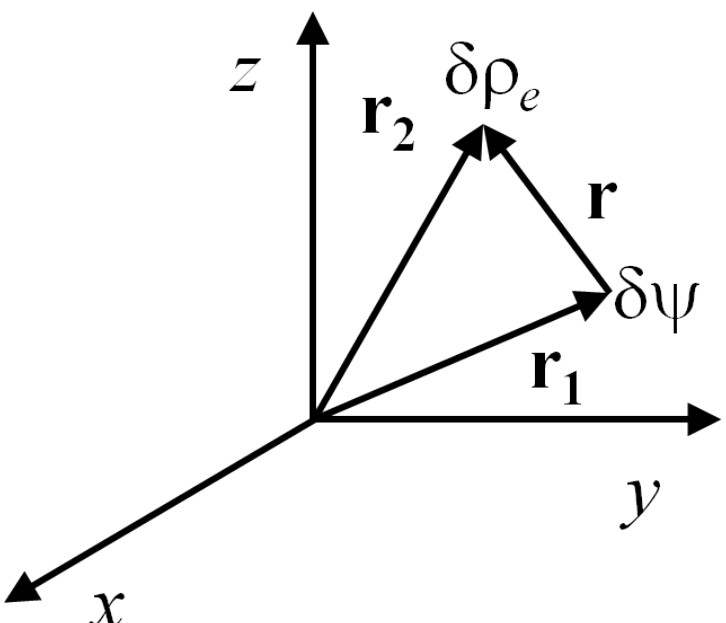

**Figure 1.** The correlation energy (50) is due to the interaction of the charge $\delta\rho_e$ with the potential $\delta\psi$.

$$\langle \delta\hat{\psi}(\mathbf{k}_1,\omega_1)\delta\hat{\rho}_e(\mathbf{k}_2,\omega_2) \rangle = -\frac{4(ze)^2cD\mathbf{k}_1 \cdot \mathbf{k}_2\delta(\omega_1+\omega_2)\delta(\mathbf{k}_1+\mathbf{k}_2)}{\varepsilon\varepsilon_0k_1^2[i\omega_1 + D(k_1^2 + \kappa^2)][i\omega_2 + D(k_2^2 + \kappa^2)]}. \tag{47}$$

Assuming $t_1 = t_2 = t$, and inverting (47) with respect to $\omega_1$ and $\mathbf{k}_1$ (see Appendix B) leads to the following result for the electrostatic energy (45)

$$\frac{E_e}{V} = \frac{k_B T\kappa^2}{8\pi}\frac{e^{-\kappa r}}{r} \tag{48}$$

Expanding the exponential leads to

$$\frac{E_e}{V} = \frac{k_B T \kappa^2}{8\pi}\left(\frac{1}{r} - \kappa + \dots\right) \approx \frac{k_B T \kappa^2}{8\pi r} - \frac{k_B T \kappa^3}{8\pi} \tag{49}$$

The first term on the right-hand side is the self-energy of the ions, while the second term accounts for the electrostatic correlation energy [14]

$$\frac{E_{corr}}{V} = -\frac{k_B T \kappa^3}{8\pi} \tag{50}$$

### 3.3. Thermodynamic Relationships
### 3.3.1. Free Energy and Chemical Potential

The focus of this section is on the effect of electrolyte solution non-ideality, which is represented by the correlation energy (50), on its thermodynamic properties. We start with the general thermodynamic relationship between the internal energy $E$ and the Helmholtz free energy $A$ [12,14]

$$\frac{E}{T^2} = -\left[\frac{\partial}{\partial T}\left(\frac{A}{T}\right)\right]_V \tag{51}$$

Substituting the correlation internal energy (50) into the above equation and integrating over the temperature yields

$$A - A_{id} = -\frac{(ze)^3 V}{12\pi\sqrt{k_B T}}\left(\frac{2c}{\varepsilon\varepsilon_0}\right)^{3/2}, \quad c = \frac{N_e}{V} \tag{52}$$

Let $N_e$ be the number of pairs of positive and negative ions, and $N_w$ is the total number of water molecules. Since we are considering a dilute electrolyte solution, it is reasonable to assume that the total volume is $V \approx N_w v_w$, where $v_w$ is the molecular volume of water. Then, Equation (52) becomes

$$A - A_{id} = -\frac{(ze)^3 (2N_e)^{3/2}}{3\sqrt{2}\pi\sqrt{k_B T}(N_w v_w)^{1/2}(\varepsilon\varepsilon_0)^{3/2}}. \tag{53}$$

Differentiating Equation (53) with respect to $N_w$ allows obtaining the chemical potential of the solvent (i.e., the water)

$$\mu_w = \left(\frac{\partial A}{\partial N_w}\right)_{N_e} - \left(\frac{\partial A_{id}}{\partial N_w}\right)_{N_e} = -\frac{\partial}{\partial N_w}\left[\frac{(ze)^3 (2N_e)^{3/2}}{3\sqrt{2}\pi\sqrt{k_B T}(N_w v_w)^{1/2}(\varepsilon\varepsilon_0)^{3/2}}\right]. \tag{54}$$

After some rearrangements, we finally obtain

$$\mu_w - \mu_w^{id} = \frac{k_B T v_w \kappa^3}{24\pi} \tag{55}$$

where $\mu_w$ is the total chemical potential and $\mu_w^{id}$ is the ideal part defined by [12,14]

$$\mu_w^{id} = \mu_w^0 + p v_w + k_B T \ln\left(\frac{N_w}{N_w + 2N_e}\right). \tag{56}$$

$\mu_w^0$ is the standard chemical potential.

### 3.3.2. Osmotic Pressure

The osmotic pressure difference between the electrolyte solution and the pure water solvent can be derived from the chemical equilibrium relationship

$$\mu_w^0 + p_0 v_w = \mu_w^0 + p v_w + k_B T \ln\left(\frac{N_w}{N_w + 2N_e}\right) + \frac{k_B T v_w \kappa^3}{24\pi} \tag{57}$$

Rearranging (57) yields

$$\Delta p = p - p_0 = -\frac{k_B T}{v_w} \ln\left(1 - \frac{2N_e}{N_w + 2N_e}\right) - \frac{k_B T \kappa^3}{24\pi} \tag{58}$$

where $\Delta p$ is the osmotic pressure of the electrolyte solution. For dilute electrolytes, $2N_e/(N_w + 2N_e) \ll 1$ and (58) simplifies to

$$\begin{aligned}\Delta p = p - p_0 &\approx \frac{k_B T}{v_w}\left(\frac{2N_e}{N_w + 2N_e}\right) - \frac{k_B T \kappa^3}{24\pi} \\ &\approx k_B T\left(\frac{2N_e}{N_w v_w}\right) - \frac{k_B T \kappa^3}{24\pi} = 2c k_B T - \frac{k_B T \kappa^3}{24\pi}.\end{aligned} \tag{59}$$

The first term on the right-hand side of Equation (59) corresponds to the ideal part of the solution osmotic pressure, while the second term accounts for the charge correlation effects. This result for the osmotic pressure also follows from the traditional Debye–Huckel approach [12,14].

### 3.3.3. Radial Distribution Functions and Pair Interaction Energies

The starting point for determining the radial distribution functions are the concentration-fluctuation correlation functions $\langle \delta c_m \mathbf{r}_+, t) \delta c_n(\mathbf{r}_+, t) \rangle$, $\langle \delta c_m \mathbf{r}_-, t) \delta c_n(\mathbf{r}_-, t) \rangle$, and $\langle \delta c_m \mathbf{r}_+, t) \delta c_n(\mathbf{r}_-, t) \rangle$, where the indices $+$ and $-$ correspond to the two ionic species (positive and negative) in the solution. Combining Equations (29) and (30) leads to

$$\begin{aligned}\frac{\partial \delta c_+}{\partial t} &= D\left(\nabla^2 \delta c_+ - \frac{\kappa}{2ze}\delta \rho_e\right) - \nabla \cdot \delta \mathbf{j}_+ \\ \frac{\partial \delta c_-}{\partial t} &= D\left(\nabla^2 \delta c_- + \frac{\kappa}{2ze}\delta \rho_e\right) - \nabla \cdot \delta \mathbf{j}_-.\end{aligned} \tag{60}$$

Transforming Equation (60) and solving for the concentration fluctuation yields

$$\begin{aligned}\delta \hat{c}_+(\mathbf{k}, \omega) &= \frac{i\mathbf{k} \cdot \delta \hat{\mathbf{j}}_+(\mathbf{k}, \omega)}{i\omega + Dk^2} - \frac{D\kappa^2}{2ze}\frac{\delta \hat{\rho}_e(\mathbf{k}, \omega)}{(i\omega + Dk^2)} \\ \delta \hat{c}_-(\mathbf{k}, \omega) &= \frac{i\mathbf{k} \cdot \delta \hat{\mathbf{j}}_-(\mathbf{k}, \omega)}{i\omega + Dk^2} + \frac{D\kappa^2}{2ze}\frac{\delta \hat{\rho}_e(\mathbf{k}, \omega)}{(i\omega + Dk^2)}\end{aligned} \tag{61}$$

Equation (61) can be used to find the Fourier transform if the concentration-fluctuations correlations for positive–positive negative–negative and positive–negative ionic combinations. These are then used to obtain the radial distribution functions and the potentials of mean force for all ionic interactions in the mean field (i.e., Debye–Huckel) limit (see also Ref. [8]).

### 3.4. Positive–Positive Ionic Correlations

The Fourier transform of the positive–positive ions concentrations fluctuation correlation is derived from the first Equation (61). After averaging, the result is

$$
\begin{aligned}
\langle \delta \hat{c}_+(\mathbf{k}_1, \omega_1) \delta \hat{c}_+(\mathbf{k}_2, \omega_2) \rangle = \\
- \frac{\mathbf{k}_1 \cdot \langle \delta \hat{\mathbf{j}}_+(\mathbf{k_1}, \omega_1) \delta \hat{\mathbf{j}}_+(\mathbf{k_2}, \omega_2) \rangle \cdot \mathbf{k}_2}{(i\omega_1 + Dk_1^2)(i\omega_2 + Dk_2^2)} + \frac{D^2 \kappa^4}{4(ze)^2} \frac{\langle \delta \hat{\rho}_e(\mathbf{k}_1, \omega_1) \delta \hat{\rho}_e(\mathbf{k}_2, \omega_2) \rangle}{(i\omega_1 + Dk_1^2)(i\omega_2 + Dk_2^2)} \\
- \frac{D\kappa^2}{2ze} \left[ \frac{i\mathbf{k}_1 \cdot \langle \delta \hat{\mathbf{j}}_+(\mathbf{k_1}, \omega_1) \delta \hat{\rho}_e(\mathbf{k}_2, \omega_2) \rangle}{(i\omega_1 + Dk_1^2)(i\omega_2 + Dk_2^2)} + \frac{i\mathbf{k}_2 \cdot \langle \delta \hat{\mathbf{j}}_+(\mathbf{k_2}, \omega_2) \delta \hat{\rho}_e(\mathbf{k}_1, \omega_1) \rangle}{(i\omega_1 + Dk_1^2)(i\omega_2 + Dk_2^2)} \right],
\end{aligned}
\tag{62}
$$

Inverting Equation (62) (see Appendices B and C) and setting $t_1 = t_2 = t$ leads to the following result for the concentration-fluctuation correlation in real space

$$
\langle \delta c_+ \mathbf{r}_1, t) \delta c_+(\mathbf{r}_2, t) \rangle = c \delta(\mathbf{r}_1 - \mathbf{r}_2) + c^2 [g_{++}(r) - 1].
\tag{63}
$$

where

$$
g_{++}(r) = 1 - \frac{(ze)^2}{4\pi\varepsilon\varepsilon_0 k_B T} \frac{e^{-\kappa r}}{r}
\tag{64}
$$

is the radial distribution function for the positive ions. Comparing (64) to the formal definition

$$
g_{++}(r) = e^{-\frac{w_{++}(r)}{k_B T}} \approx 1 - \frac{w_{++}(r)}{k_B T}
\tag{65}
$$

leads to the Debye and Huckel [11] result for the electrostatic energy of interaction between the ions in the solution

$$
w_{++}(r) = ze\psi(r) = \frac{(ze)^2}{4\pi\varepsilon\varepsilon_0} \frac{e^{-\kappa r}}{r}.
\tag{66}
$$

The energy is positive, which indicates a repulsive interaction.

### 3.5. Negative–Negative Ionic Correlations

Using the second Equation (61), we obtain the concentration negative–negative ion concentration-fluctuation correlation in the form

$$
\begin{aligned}
\langle \delta \hat{c}_-(\mathbf{k}_1, \omega_1) \delta \hat{c}_-(\mathbf{k}_2, \omega_2) \rangle = \\
- \frac{\mathbf{k}_1 \cdot \langle \delta \hat{\mathbf{j}}_-(\mathbf{k_1}, \omega_1) \delta \hat{\mathbf{j}}_-(\mathbf{k_2}, \omega_2) \rangle \cdot \mathbf{k}_2}{(i\omega_1 + Dk_1^2)(i\omega_2 + Dk_2^2)} + \frac{D^2 \kappa^4}{4(ze)^2} \frac{\langle \delta \hat{\rho}_e(\mathbf{k}_1, \omega_1) \delta \hat{\rho}_e(\mathbf{k}_2, \omega_2) \rangle}{(i\omega_1 + Dk_1^2)(i\omega_2 + Dk_2^2)} \\
+ \frac{D\kappa^2}{2ze} \left[ \frac{i\mathbf{k}_1 \cdot \langle \delta \hat{\mathbf{j}}_-(\mathbf{k_1}, \omega_1) \delta \hat{\rho}_e(\mathbf{k}_2, \omega_2) \rangle}{(i\omega_1 + Dk_1^2)(i\omega_2 + Dk_2^2)} + \frac{i\mathbf{k}_2 \cdot \langle \delta \hat{\mathbf{j}}_-(\mathbf{k_2}, \omega_2) \delta \hat{\rho}_e(\mathbf{k}_1, \omega_1) \rangle}{(i\omega_1 + Dk_1^2)(i\omega_2 + Dk_2^2)} \right].
\end{aligned}
\tag{67}
$$

Note that the sign in front of the last two terms on the right is positive, which is in contrast to Equation (62). Following the same recipe as for the positive–positive ion interactions above (see Appendix C), we derive

$$
\langle \delta c_-(\mathbf{r}_1, t) \delta c_-(\mathbf{r}_2, t) \rangle = c \delta(\mathbf{r}_1 - \mathbf{r}_2) + c^2 [g_{--}(r) - 1].
\tag{68}
$$

The radial distribution function is

$$
g_{--}(r) = 1 - \frac{(ze)^2}{4\pi\varepsilon\varepsilon_0 k_B T} \frac{e^{-\kappa r}}{r},
\tag{69}
$$

and the interaction energy reads [11]

$$
w_{--}(r) = \frac{(ze)^2}{4\pi\varepsilon\varepsilon_0} \frac{e^{-\kappa r}}{r},
\tag{70}
$$

which is also positive (i.e., repulsive). Hence, the radial distribution functions and interaction energies for ions that are both positive or both negative are identical.

*3.6. Positive–Negative Ionic Correlations*

The positive–negative ion concentration-fluctuation correlation is also derived from Equation (61). The result becomes

$$
\langle \delta \hat{c}_+(\mathbf{k}_1, \omega_1) \delta \hat{c}_-(\mathbf{k}_2, \omega_2) \rangle = -\frac{D^2 \kappa^4}{4(ze)^2} \frac{\langle \delta \hat{\rho}_e(\mathbf{k}_1, \omega_1) \delta \hat{\rho}_e(\mathbf{k}_2, \omega_2) \rangle}{(i\omega_1 + Dk_1^2)(i\omega_2 + Dk_2^2)}
$$
$$
+ \frac{D\kappa^2}{2ze} \left[ \frac{i\mathbf{k}_1 \cdot \langle \delta \hat{\mathbf{j}}_+(\mathbf{k}_1, \omega_1) \delta \hat{\rho}_e(\mathbf{k}_2, \omega_2) \rangle}{(i\omega_1 + Dk_1^2)(i\omega_2 + Dk_2^2)} - \frac{i\mathbf{k}_2 \cdot \langle \delta \hat{\mathbf{j}}_-(\mathbf{k}_2, \omega_2) \delta \hat{\rho}_e(\mathbf{k}_1, \omega_1) \rangle}{(i\omega_1 + Dk_1^2)(i\omega_2 + Dk_2^2)} \right]. \tag{71}
$$

The correlation between the positive and negative ionic fluxes is zero because of Equation (39). In addition, $\langle \delta \hat{c}_+(\mathbf{k}_1, \omega_1) \delta \hat{c}_-(\mathbf{k}_2, \omega_2) \rangle = \langle \delta \hat{c}_-(\mathbf{k}_1, \omega_1) \delta \hat{c}_+(\mathbf{k}_2, \omega_2) \rangle$. The inversion of Equation (71) at $t_1 = t_2 = t$ yields

$$
\langle \delta c_+(\mathbf{r}_1, t) \delta c_-(\mathbf{r}_1, t) \rangle = c^2 [g_{+-}(r) - 1]. \tag{72}
$$

The radial distribution function and pair interaction energy between oppositely charged ions are

$$
g_{+-}(r) = 1 + \frac{(ze)^2}{4\pi\varepsilon\varepsilon_0 k_B T} \frac{e^{-\kappa r}}{r}, \tag{73}
$$

and [11]

$$
w_{+-}(r) = -\frac{(ze)^2}{4\pi\varepsilon\varepsilon_0} \frac{e^{-\kappa r}}{r}. \tag{74}
$$

The sign of the interaction energy $w_{+-}(r)$ is negative, which is in agreement with the fact that positive and negative ions attract each other.

## 4. Conclusions

The Landau–Lifshitz approach for the treatment of hydrodynamic fluctuations is based on the fundamental expression for entropy production (4). Its generalization to include transport processes (such as diffusion) is straightforward. The method offers simple expressions for the correlation of the fluctuating fluxes. The flux fluctuations are related to the concentration fluctuations, which in turn account for the effects of interactions. Proper averaging allows us to derive results that are pertinent to systems in thermodynamic equilibrium. The procedure is relatively simple, and the main mathematical techniques are the forward and inverse Fourier transforms, and more specifically the transforms of various Dirac delta functions.

The focus of this paper is on electrolyte solutions. We show how the Landau and Lifshitz method can be used to derive the Debye and Huckel theory of strong, dilute electrolytes. This is accomplished without formally solving the Poisson equation of electrostatics (25). Even the Debye and Huckel assumption of low potential and the subsequent linearization of the Boltzmann distribution of the local charge

$$
-\frac{\rho(\mathbf{r})}{\epsilon\epsilon_0} = 2zec \sinh\left[\frac{ze\psi(\mathbf{r})}{k_B T}\right] \approx \frac{2(ze)^2 c}{\epsilon\epsilon_0 k_B T} \psi = \kappa^2 \psi \tag{75}
$$

is not explicitly applied. Instead, we use the relationship between charge and potential fluctuations (30) to write the transport Equation (29) in terms of concentration fluctuations. The equilibrium state, in our analysis, is established by the integration (averaging) over the entire frequency spectrum in Fourier space. The assumptions $\delta c_+ \ll c$, $\delta c_- \ll c$, $\delta \rho_e \ll \rho_e$, and $\delta \psi \ll \psi$ are equivalent to the Debye and Huckel low potential approximation $ze\psi(\mathbf{r})/k_B T < 1$. However, the fluctuation correlation analysis clearly demonstrates that the low potential approximation is implied by the small charge density fluctuations, which

are a direct consequence of the small concentration fluctuations. This is also true for the Debye and Huckel analysis [11], although there, the argument starts with the low potential approximation. Another curious feature of the method is that although it is examining the fluctuations in the system, the end result is a mean field theory, which is analogous to the Debye and Huckel result.

The correlation energy (45) and the osmotic pressure, Equations (58) and (59), are entirely determined by the charge density fluctuation correlations given by Equations (43) and (44). The pair energy of interaction positive–positive, negative–negative, and neagtive–positive ions depends on the charge density fluctuation correlations but also includes contributions from the fluctuation correlations of respective ionic fluxes with those of the charge density (see Equations (62), (67) and (71)). The terms that are proportional to the positive–positive and negative–negative flux fluctuation correlations in (62) and (67) do not contribute at all to the pair interaction energy but are responsible for the $c\delta(\mathbf{r}_1 - \mathbf{r}_2)$ term in Equations (62) and (67). There is no such term in Equation (71) because the diffusion fluxes for the positive and negative ions are independent and uncorrelated (see Equations (39) and (40)).

The assumption that all diffusion coefficients are the same ($D_+ = D_- = D$) can be abandoned, which makes all derivations and calculations significantly more lengthy and tedious. The final results, however, for the osmotic pressure, interaction energies and radial distribution functions are not affected. This is not surprising, since thermodynamic properties cannot depend on dissipative coefficients such as the diffusivities.

**Funding:** This research received no external funding.

**Acknowledgments:** The author is indebted to the late Peter Kralchevsky for the many illuminating discussions of these topics.

**Conflicts of Interest:** The authors declare no conflict of interest.

## Abbreviations

| | |
|---|---|
| $A$ | Helmholtz free energy |
| $c$ | concentration of dissolved species |
| $D$ | diffusion coefficient |
| $E$ | internal energy |
| $E_{corr}$ | correlation energy |
| $E_e$ | electrostatic energy |
| $e$ | elementary charge |
| $g$ | radial distribution function |
| $h$ | total correlation function |
| $\mathbf{I}$ | unit tensor |
| $i, \mathbf{i}$ | flux |
| $j, \mathbf{j}$ | flux per unit mass |
| $k, \mathbf{k}$ | wave number and wave vector |
| $k_B$ | Boltzmann constant |
| $m$ | mass of dissolved species |
| $N$ | number of dissolved species |
| $N_e$ | number of ion pairs |
| $N_w$ | number of water molecules |
| $p$ | pressure |
| $\mathbf{r}$ | spatial coordinate (radius-vector) |
| $S$ | entropy |
| $T$ | temperature |
| $t$ | time |
| $V$ | volume |

| | |
|---|---|
| $w$ | potential of mean force |
| $X$ | generalized thermodynamic force |
| $\dot{x}$ | generalized flux |
| $y$ | generalized fluctuating flux |
| $z$ | ionic charge number |
| $\alpha$ | transport coefficient |
| $\beta$ | friction coefficient |
| $\gamma$ | Onsager kinetic coefficient |
| $\delta$ | coefficient, indicating a fluctuating quantity |
| $\delta(x)$ | delta function ($x$ is an arbitrary variable) |
| $\delta_{\eta\zeta}$ | Kronecker symbol |
| $\Delta V$ | small but finite volume |
| $\varepsilon$ | dielectric permittivity |
| $\varepsilon_0$ | dielectric constant *in vacuo* |
| $\zeta$ | coordinate index |
| $\eta$ | coordinate index |
| $\kappa$ | Debye screening parameter |
| $\mu$ | chemical potential |
| $\psi$ | electrostatic potential |
| $\omega$ | frequency |

## Appendix A. Fourier Transform of the Mass Fluxes Fluctuation Correlations

Here, we derive the Fourier transform of the fluctuating flux correlation (23), or

$$
\langle \delta\hat{\mathbf{j}}(\mathbf{k}_1, \omega_1) \delta\hat{\mathbf{j}}(\mathbf{k}_2, \omega_2) \rangle =
$$
$$
\frac{1}{(2\pi)^4} \int_{-\infty}^{\infty} d\mathbf{r}_1 \int_{-\infty}^{\infty} dt_1 \int_{-\infty}^{\infty} d\mathbf{r}_2 \int_{-\infty}^{\infty} dt_2 \langle \delta\mathbf{j}(\mathbf{r}_1, t_1)\delta\mathbf{j}(\mathbf{r}_2, t_2)\rangle e^{i(\omega_1 t_1 - \mathbf{k}_1 \cdot \mathbf{r}_1)} e^{i(\omega_2 t_2 - \mathbf{k}_2 \cdot \mathbf{r}_2)}. \quad \text{(A1)}
$$

Hence, all that is needed is to transform two delta functions $\delta(t_1 - t_2)$ and $\delta(\mathbf{r_1} - \mathbf{r_2})$ over the two times ($t_1$ and $t_2$) and positions ($\mathbf{r_1}$ and $\mathbf{r_2}$) variables, or [16]

$$
\frac{1}{(2\pi)^4} \int_{-\infty}^{\infty} d\mathbf{r}_1 \int_{-\infty}^{\infty} dt_1 \int_{-\infty}^{\infty} d\mathbf{r}_2 \int_{-\infty}^{\infty} dt_2 [2c\mathbf{I}\delta(t_1 - t_2)\delta(\mathbf{r}_1 - \mathbf{r}_2)] \times
$$
$$
e^{i(\omega_1 t_1 - \mathbf{k_1}\cdot\mathbf{r_1})} e^{i(\omega_2 t_2 - \mathbf{k_2}\cdot\mathbf{r_2})} = 2c\mathbf{I}D\delta(\omega_1 + \omega_2)\delta(\mathbf{k}_1 + \mathbf{k}_2). \quad \text{(A2)}
$$

The mass fluxes fluctuation correlation result (A2) is used to find the charge–potential fluctuation correlations and ionic concentration-fluctuation correlations as shown below.

## Appendix B. Electrostatic Charge–Potential Correlations and the Electrolyte Contribution to the Osmotic Pressure

Below, we provide details on the Fourier inversion of the potential–charge correlation expression (47) with respect to $\omega_1$ and $\mathbf{k_1}$. Starting with the frequency, we write

$$
\frac{1}{(2\pi)^{1/2}} \int_{-\infty}^{\infty} d\omega_1 \langle \delta\hat{\psi}(\mathbf{k}_1, \omega_1)\delta\hat{\rho}_e(\mathbf{k}_2, \omega_2)\rangle e^{-it(\omega_1 + \omega_2)} =
$$
$$
-\int_{-\infty}^{\infty} d\omega_1 \frac{4(ze)^2 cD\mathbf{k}_1 \cdot \mathbf{k}_2 \delta(\omega_1 + \omega_2)\delta(\mathbf{k}_1 + \mathbf{k}_2)e^{-it(\omega_1 + \omega_2)}}{(2\pi)^{1/2}\varepsilon\varepsilon_0 k_1^2 [i\omega_1 + D(k_1^2 + \kappa^2)][i\omega_2 + D(k_2^2 + \kappa^2)]}. \quad \text{(A3)}
$$

The right-hand side can be further rearranged to read

$$
-\frac{4(ze)^2 cD\mathbf{k}_1 \cdot \mathbf{k}_2}{(2\pi)^{1/2}\varepsilon\varepsilon_0 k_1^2} \int_{-\infty}^{\infty} d\omega_1 \frac{\delta(\omega_1 + \omega_2)\delta(\mathbf{k}_1 + \mathbf{k}_2)e^{-it(\omega_1 + \omega_2)}}{[i\omega_1 + D(k_1^2 + \kappa^2)][i\omega_2 + D(k_2^2 + \kappa^2)]} =
$$
$$
\frac{4(ze)^2 cD\mathbf{k}_1 \cdot \mathbf{k_2}}{(2\pi)^{1/2}\varepsilon\varepsilon_0 k_1^2} \frac{\delta(\mathbf{k}_1 + \mathbf{k}_2)}{[i\omega_2 - D(k_1^2 + \kappa^2)][i\omega_2 + D(k_2^2 + \kappa^2)]}. \quad \text{(A4)}
$$

Next, we perform the inverse Fourier transform over the wave vectors starting with $\mathbf{k}_1$

$$
\begin{aligned}
&\frac{4(ze)^2cD}{(2\pi)^2\varepsilon\varepsilon_0}\int_{-\infty}^{\infty}d\mathbf{k}_1\frac{\mathbf{k}_1\cdot\mathbf{k}_2\delta(\mathbf{k}_1+\mathbf{k}_2)e^{-i(\mathbf{k}_1\cdot\mathbf{r}_1-\mathbf{k}_2\cdot\mathbf{r}_2)}}{k_1^2[i\omega_2-D(k_1^2+\kappa^2)][i\omega_2+D(k_2^2+\kappa^2)]}=\\
&\frac{4(ze)^2cD}{(2\pi)^2\varepsilon\varepsilon_0}\frac{e^{i\mathbf{k}_2\cdot\Delta\mathbf{r}}}{[\omega_2^2+D^2(k_2^2+\kappa^2)^2]}.
\end{aligned}
\tag{A5}
$$

where $\Delta\mathbf{r}=\mathbf{r}_2-\mathbf{r}_1$. Now, we can repeat the same procedure over the frequency $\omega_2$

$$
\frac{4(ze)^2cDe^{i\mathbf{k}_2\cdot\delta\mathbf{r}}}{(2\pi)^{5/2}\varepsilon\varepsilon_0}\int_{\infty}^{\infty}\frac{d\omega_2}{[\omega_2^2+D^2(k_2^2+\kappa^2)^2]}=\frac{4(ze)^2ce^{i\mathbf{k}_2\cdot\Delta\mathbf{r}}}{(2\pi)^{5/2}\varepsilon\varepsilon_0}\frac{\pi}{k_2^2+\kappa^2}.
\tag{A6}
$$

Finally, we integrate over the wave vector $\mathbf{k}_2$, which leads to

$$
\langle\delta\psi(\mathbf{r}_1,t_1)\delta\rho_e(\mathbf{r}_2,t_2)\rangle=\frac{4\pi(ze)^2c}{(2\pi)^4\varepsilon\varepsilon_0}\int_{-\infty}^{\infty}d\mathbf{k}_2\frac{e^{i\mathbf{k}_2\cdot\Delta\mathbf{r}}}{k_2^2+\kappa^2}
\tag{A7}
$$

This is easy to accomplish if we select a coordinate system where the vector $\Delta\mathbf{r}$ is directed along $k_{2z}$ axis (see Figure A1). Then, $\mathbf{k}_2\cdot\Delta\mathbf{r}=k_2r\cos\theta$ and $d\mathbf{k}_2=k_2^2\sin\theta d\theta d\phi$, and the integral in (A7) becomes

$$
\begin{aligned}
&\int_{-\infty}^{\infty}d\mathbf{k}_2\frac{e^{i\mathbf{k}_2\cdot\Delta\mathbf{r}}}{k_2^2+\kappa^2}=\int_0^{\infty}\int_0^{\pi}\int_0^{2\pi}dk_2d\theta d\phi\frac{k_2^2\sin\theta e^{ikr\cos\theta}}{k_2^2+\kappa^2}=\\
&2\pi\int_0^{\infty}dk_2\frac{k_2^2}{k_2^2+\kappa^2}\frac{e^{ikr}-e^{-ikr}}{ikr}=\frac{4\pi}{r}\int_0^{\infty}dk_2\frac{k_2\sin(k_2r)}{k_2^2+\kappa^2}=2\pi^2\frac{e^{-\kappa r}}{r}
\end{aligned}
\tag{A8}
$$

Inserting (A8) in Equation (A7) leads to the final results for the potential–charge correlation (see Equation (33)).

$$
\langle\delta\psi(\mathbf{r}_1,t)\delta\rho(\mathbf{r}_2,t)\rangle=\frac{k_BT\kappa^2}{4\pi}\frac{e^{-\kappa r}}{r}
\tag{A9}
$$

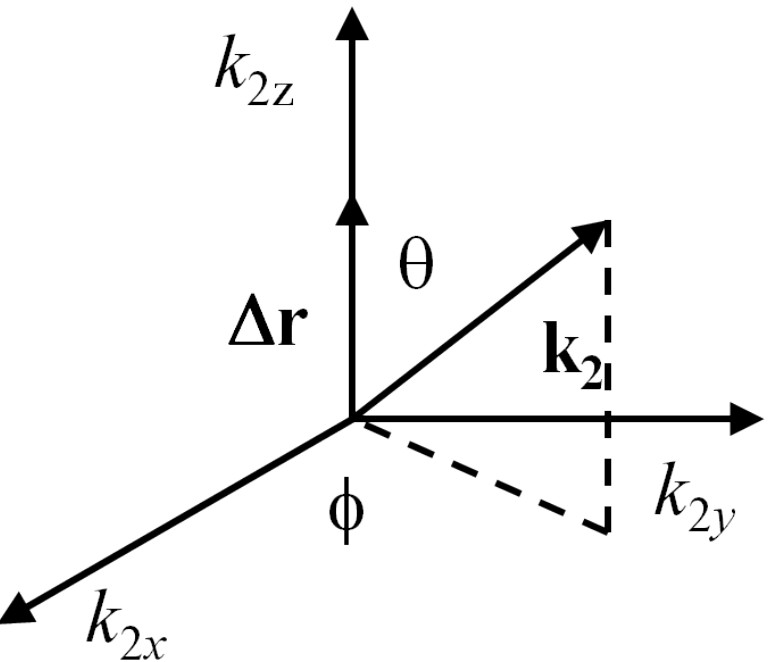

**Figure A1.** Coordinate setup for calculating the integral (A8). $\theta$ and $\phi$ are the polar and azimuth angles, and $k_2^2=k_{2x}^2+k_{2y}^2+k_{2z}^2$.

## Appendix C. Derivation of Ionic Concentration Correlations

This section provides details on the derivation of the concentration fluctuations (63), (68) and (72), starting from (62), (67) and (71). The necessary mathematical procedure is the inverse Fourier transform, which we will perform separately on each term for better clarity.

### *Appendix C.1. Flux Correlation Term*

Starting with the first terms in Equations (63) and (68), we obtain (taking the integrals one at the time as in the previous section)

$$
-\frac{1}{(2\pi)^4} \int_{-\infty}^{\infty} d\mathbf{k}_1 \int_{-\infty}^{\infty} d\mathbf{k}_2 \int_{-\infty}^{\infty} d\omega_1 \int_{-\infty}^{\infty} d\omega_2 \left[ \frac{\mathbf{k}_1 \cdot \langle \delta\hat{\mathbf{j}}_{\pm}(\mathbf{k}_1,\omega_1)\delta\hat{\mathbf{j}}_{\pm}(\mathbf{k}_2,\omega_2)\rangle \cdot \mathbf{k}_2}{(i\omega_1 + Dk_1^2)(i\omega_2 + Dk_2^2)} \right] \times
$$
$$
e^{-i(\omega_1 t_1 + \omega_2 t_2 - \mathbf{k}_1 \cdot \mathbf{r}_1 - \mathbf{k}_2 \cdot \mathbf{r}_2)} = c\delta(\mathbf{r}_1 - \mathbf{r}_2)\delta(t_1 - t_2). \tag{A10}
$$

The final result is the same for both positive or both negative ionic flux correlations. Since $\langle \delta\hat{\mathbf{j}}_{+}(\mathbf{k}_1,\omega_1)\delta\hat{\mathbf{j}}_{-}(\mathbf{k}_2,\omega_2)\rangle = \langle \delta\hat{\mathbf{j}}_{-}(\mathbf{k}_1,\omega_1)\delta\hat{\mathbf{j}}_{+}(\mathbf{k}_2,\omega_2)\rangle = 0$ (see Equation (39)), there is no such term in Equation (72).

### *Appendix C.2. Charge Correlation Term*

The next term of interest is proportional to the charge density correlation, and its inverse Fourier transform is

$$
-\frac{1}{(2\pi)^4} \int_{-\infty}^{\infty} d\mathbf{k}_1 \int_{-\infty}^{\infty} d\mathbf{k}_2 \int_{-\infty}^{\infty} d\omega_1 \int_{-\infty}^{\infty} d\omega_2 \left[ \frac{D^2\kappa^4}{4(ze)^2} \frac{\langle \delta\hat{\rho}_e(\mathbf{k}_1,\omega_1)\delta\hat{\rho}_e(\mathbf{k}_2,\omega_2)\rangle}{(i\omega_1 + Dk_1^2)(i\omega_2 + Dk_2^2)} \right] \times
$$
$$
e^{i(\omega_1 t_1 + \omega_2 t_2 - \mathbf{k}_1 \cdot \mathbf{r}_1 - \mathbf{k}_2 \cdot \mathbf{r}_2)} = \frac{c(ze)^2}{4\pi\varepsilon\varepsilon_0 k_B T} \left( \frac{e^{-\frac{\kappa}{\sqrt{2}}r}}{r} - \frac{e^{-\kappa r}}{r} \right). \tag{A11}
$$

### *Appendix C.3. Flux-Charge Correlations*

The last terms in Equations (62), (67) and (71) account for the correlations between the ionic flux (positive of negative) and the charge density fluctuations. Starting with the last term in Equation (62), and using Equations (39), (40) and (42), we obtain

$$
-\frac{D\kappa^2}{2ze} \left[ \frac{i\mathbf{k}_1 \cdot \langle \delta\hat{\mathbf{j}}_{+}(\mathbf{k}_1,\omega_1)\delta\hat{\rho}_e(\mathbf{k}_2,\omega_2)\rangle}{(i\omega_1 + Dk_1^2)(i\omega_2 + Dk_2^2)} + \frac{i\mathbf{k}_2 \cdot \langle \delta\hat{\mathbf{j}}_{+}(\mathbf{k}_2,\omega_2)\delta\hat{\rho}_e(\mathbf{k}_1,\omega_1)\rangle}{(i\omega_1 + Dk_1^2)(i\omega_2 + Dk_2^2)} \right] =
$$
$$
\left\{ \frac{\mathbf{k}_1 \cdot \mathbf{k}_2 \kappa^2 c D^2 \delta(\omega_1 + \omega_2)\delta(\mathbf{k}_1 + \mathbf{k}_2)}{(i\omega_1 + Dk_1^2)(i\omega_2 + Dk_2^2)[i\omega_2 + D(k_2^2 + \kappa^2)]} + \right.
$$
$$
\left. \frac{\mathbf{k}_1 \cdot \mathbf{k}_2 \kappa^2 c D^2 c D\delta(\omega_1 + \omega_2)\delta(\mathbf{k}_1 + \mathbf{k}_2)}{(i\omega_1 + Dk_1^2)(i\omega_2 + Dk_2^2)[i\omega_1 + D(k_1^2 + \kappa^2)]} \right\}. \tag{A12}
$$

The right-hand side above can be inverse-transformed following the procedure outlined in Appendix B above. The result is

$$
-\frac{c(ze)^2}{4\pi\varepsilon\varepsilon_0 k_B T} \frac{e^{-\frac{\kappa}{\sqrt{2}}r}}{r}. \tag{A13}
$$

Note that (A13) is identical to the first term in the right-hand side of Equation (A11) but with an opposite sign. This is also true for the cases of negative–negative and positive–

negative concentration fluctuations. They always cancel, and summing the results from Equations (A10), (A11) and (A13) leads to

$$\langle \delta c_+(\mathbf{r}_1, t) \delta c_+(\mathbf{r}_1, t) \rangle = c \left[ \delta(\mathbf{r}_1 - \mathbf{r}_2) - \frac{c(ze)^2}{4\pi\varepsilon\varepsilon_0 k_B T} \frac{e^{-\kappa r}}{r} \right], \tag{A14}$$

which is equivalent to Equation (63). Equations (68) and (72) can be derived via the same mathematical procedure to read

$$\langle \delta c_-(\mathbf{r}_1, t) \delta c_-(\mathbf{r}_1, t) \rangle = c \left[ \delta(\mathbf{r}_1 - \mathbf{r}_2) - \frac{c(ze)^2}{4\pi\varepsilon\varepsilon_0 k_B T} \frac{e^{-\kappa r}}{r} \right] \tag{A15}$$

and

$$\langle \delta c_+(\mathbf{r}_1, t) \delta c_-(\mathbf{r}_1, t) \rangle = c \left[ \delta(\mathbf{r}_1 - \mathbf{r}_2) + \frac{c(ze)^2}{4\pi\varepsilon\varepsilon_0 k_B T} \frac{e^{-\kappa r}}{r} \right]. \tag{A16}$$

The signs' electrostatic contributions to the correlations are positive for the same-charge cases and negative for the positive–negative case. This obviously reflects the fact that same-charge ions repel, while oppositely charged ions attract each other.

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
