# Peer review of "Thermodynamic Properties of Electrolyte Solutions, Derived from Fluctuation Correlations: A Methodological Review"

_applsci, doi:10.3390/app12125863_

Round 1

Reviewer 1 Report

This work presents an overview of the Landau and Lifshitz approach for analysis of hydrodynamic fluctuations and extends this approach to dilute electrolyte solutions. In particular, the author shows how this methodology reproduces results from the Debye and Huckel theory while starting from a very different physical perspective. The manuscript is clear, and the different assumptions are well justified.

In order to better understand what is new in this approach, I suggest that the author explained how formula (66) in the manuscript is obtained directly from the Debye Huckel Theory and why it is the same as the formula (64) obtained by the methodology developped in this work.

after this minor revision, I recommend this work for a publication in Applied Sciences.

Author Response

I appreciate the positive assessment of the work by the reviewer. There are two questions that need addressing, which follows below.

  1. "how formula (66) in the manuscript is obtained directly from the Debye Huckel Theory"

Eq. (66) represents the energy of interaction between two ions in solution. According to formal electrostatics, as well as from a thermodynamic argument, this energy can be expressed by the work done to bring a the two ions at a distance r starting from infinity. Let ion 1 creates a potential around itself equal to ψ(r). Then bringing a second ion 2 near ion 1 requires work equal to zeψ(r). The potential ψ(r) = Ae-κr/r was derived in the original paper by Debye and Huckel (see the discussion between Eqs. (22) and (23) in their paper. For diluted electrolytes Aze/εε0 (in SI units). That is why expression (66) in the manuscript is often referred to as the Debye and Huckel energy. 

2. "why it is the same as the formula (64) obtained by the methodology developed in this work?"

The probability of finding a second ion near the first one in equilibrium (the radial distribution function) is given by a Boltzmann type of distribution that has the interaction energy in the power. For low energies of interaction the exponential is expanded and truncated at the linear term as shown by Eq. (65). Eq. (66) is obtained by simply comparing Eqs. (64) and (65). 

To clarify this point, Eq. (66) has been rearranged. 

Reviewer 2 Report

It is unclear to me if in the end the manuscript "Thermodynamic properties of electrolyte solutions, derived from fluctuation correlations" is supposed to be a review (first line of the abstract) or an original contribution (beginning of the second paragraph int he Introduction).

In any case, the work of the author is, in my opinion, preceded by two publications dealing with the low Mach number fluctuating hydrodynamics of fluids (which is the approach by Landau & Lifshitz). Unfortunately, it seems that the author was not aware of these publications.

The first one, Phys. of Fluids 27, 037103 (2015) presents a thorough, and very detailed study of (uncharged) multispecies liquid mixtures; the sections I and II of this paper present the generalization of the analysis discussed by the author in Sec. 2 of the current manuscript.

The second one, Phys. Rev. Fluids 1, 074103 (2016) deals with electrolytes; the sections II and II of this paper cover, again in much more detail, the analysis discussed by the author in Sec. 3., including the derivation of the pair correlation function (see Eq.(49), and the discussion around it, of that reference).

Furthermore, these two rather comprehensive studies also contain a large number of references relevant to the use of the approach of fluctuating hydrodynamics in statistical physics.

Accordingly, I find the submitted manuscript unsuitable for publication.

Author Response

Response to the comments of Reviewer 2

1. "It is unclear to me if in the end the manuscript "Thermodynamic properties of electrolyte solutions, derived from fluctuation correlations" is supposed to be a review (first line of the abstract) or an original contribution (beginning of the second paragraph int he Introduction)."

It is a review. The revised version attempts to make this clearer. 

2. "In any case, the work of the author is, in my opinion, preceded by two publications dealing with the low Mach number fluctuating hydrodynamics of fluids (which is the approach by Landau & Lifshitz). Unfortunately, it seems that the author was not aware of these publications."

These publications are indeed very comprehensive. The main focus is on non-equilibrium systems and gradient driven transport, and less on the equilibrium implications. Indeed they provide some results from the Debye and Huckel theory, but without detailed explanation of the approach, which no doubt the authors are fully aware of. The references are added in the introduction as well as in the results section. 

The purpose of the present review is to advocate for the power of the method in general and to show it allows for derivation of analytical results for systems in thermodynamic equilibrium. 

Reviewer 3 Report

Manuscript Review «Thermodynamic Properties of Electrolyte Solutions, Derived from Fluctuation Correlations»

The manuscript considers the problem of fluctuating transport flows in hydrodynamics. The method is related to fluctuation concentrations, molecular interactions and correlation of fluctuation charge strength. The energy of the ionic vapor is taken into account, which depends on the fluctuations of the charge force. The contributions of the fluctuation correlation of the corresponding ion fluxes with different values ​​of charge power are also taken into account. The approach is relevant for use in calculations of thermodynamic processes, such as osmotic pressure and the interaction between soluble phases.

DISADVANTAGES - the manuscript can be improved by supporting the calculations with illustrative material (in the form of graphic dependencies, at least 2-3), in particular in paragraphs 3.3.2 and 3.3.3

CONCLUSION: The manuscript may be published after a minor correction.

Author Response

We appreciate the positive assessment of the our work, expressed by the reviewer. There is one critical comment reads:

"DISADVANTAGES - the manuscript can be improved by supporting the calculations with illustrative material (in the form of graphic dependencies, at least 2-3), in particular in paragraphs 3.3.2 and 3.3.3"

The results in sections 3.3.2 and 3.3.3 are the osmotic pressure and radial distribution functions expressions. They are not new and in fact well-known. They were also not the the main goal of this work. It is their derivation, using the Landau&Lifshitz method that is the point of this review. In other words are advocating in favor of this approach and the derivation of these formulas is just  to demonstrate the power of the method. It would be awkward to plot figures that are known from the last century and have found their way into many texts on the subject. 

The two figures that are already in the text were essential to explain the ideal behind the approach, and how exactly the above results were derived. 

Reviewer 4 Report

The manuscript describes thermodynamic properties of electrolyte solutions based on Landau-Lifshitz approach for treatment of hydrodynamic fluctuations. The author used Fourier transform for correlation of the fluctuating fluxes and made the procedure relatively simple. Also, the author shows the correlation energy, the osmotic pressure in terms of the relationship between charge and potential fluctuations at electrolyte to write the transport equations. However, there are some parts that need to be fixed a little more. Therefore, I recommend this manuscript be published after a minor revision.

Comment 1 :

Typos and grammatical errors are frequently found throughout the manuscript. As an example, the following errors were found (Page 1 line 25 (Nernts), Page 3 and Page 4 line 47 (the the), Page 4 line 52 (desccribed), Page 5 (is does), Page 11 line 109 (do no)). It is recommended that the author take English proofreading before final publication.

Comment 2 :

There are many Greek symbols throughout the paper to elicit the author's arguments. It would be better to make a nomenclature table of symbols to clarify the reader's understanding.

Author Response

The overall positive assessment of the manuscript is greatly appreciated. Below are the responses to the comments

  1. "Typos and grammatical errors are frequently found throughout the manuscript. As an example, the following errors were found (Page 1 line 25 (Nernts), Page 3 and Page 4 line 47 (the the), Page 4 line 52 (desccribed), Page 5 (is does), Page 11 line 109 (do no)). It is recommended that the author take English proofreading before final publication."

I would like to thank the reviewer for pointing to the presence of typos in the text. They are now fixed. 

2. "There are many Greek symbols throughout the paper to elicit the author's arguments. It would be better to make a nomenclature table of symbols to clarify the reader's understanding."

A list of symbols is now added to the manuscript. 

Reviewer 5 Report

  1. In the abstract, references are usually not recommended, so authors should remove the references.
  2. They said in the abstract, “this article presents the review…...” so please state either its review or research article.
  3. If this is a review, the authors improve the introduction section by reviewing more literature.
  4. The authors put a lot of mature equations, but they should put some literature or studies about the applications of these equations.
  5. The authors should mention the recent applications of these equations.
  6. Authors should put recent references in the reference section.

Author Response

The responses to reviewer 5 are below.

1. "In the abstract, references are usually not recommended, so authors should remove the references."

The references are removed.

2. "They said in the abstract, “this article presents the review…...” so please state either its review or research article."

It is a review, and it was submitted as such. The text was changed to emphasize that. 

3. "If this is a review, the authors improve the introduction section by reviewing more literature."

The point of the review is to show how Landau&Lifshitz method can be used to obtain simple analytical results beyond just hydrodynamics. There are not many papers on that, which is part of the motivation to write this review, namely to encourage the use of the method for other applications. Still a few references were added, including one that is very close and recent, but its main (and much broader) focus is on a numerical instead of an analytical treatment instead of similar systems. A discussion of those works is also added to the text. 

4. "The authors put a lot of mature equations, but they should put some literature or studies about the applications of these equations."

See answer to 3.

5. "The authors should mention the recent applications of these equations."

See answer to 3. 

6. "Authors should put recent references in the reference section."

See answer to 3. 

Round 2

Reviewer 2 Report

I thank the author for the reply; I now understand that the manuscript is a review. However, I am not convinced that the aspects noted and discussed by the author justify the publication of a brief "review".

Author Response

The reviewer is certainly entitled to his/hers opinion, which I do not share. 

Reviewer 5 Report

According to the authors, this paper is a review paper, but the authors put only 16 references, and all references are old. The authors replied (below) that not many papers on this topic. So my specific comment is as follows:

   1. They should put the more recent references to improve the review.

    2. They should put a detailed explanation of the approach.

Author Response

There are two comments by the reviewer that are addressed below.

 1. They should put the more recent references to improve the review.

There are newer references already added to the revised version of manuscript (Refs. 7 and 8). 

2. They should put a detailed explanation of the approach.

The whole manuscript is literally a detailed step by step instruction how to use the approach.